# Role of Human Microbiome in Development and Management of Head and Neck Squamous Cell Carcinoma

**DOI:** 10.3390/cancers17132238

**Published:** 2025-07-03

**Authors:** Martin Palkovsky, Nikol Modrackova, Vera Neuzil-Bunesova, Marian Liberko, Alzbeta Hlodakova, Renata Soumarova

**Affiliations:** 1Department of Oncology, University Hospital Kralovske Vinohrady, 100 34 Prague, Czech Republic; marian.liberko@fnkv.cz (M.L.); alzbeta.hlodakova@fnkv.cz (A.H.); renata.soumarova@fnkv.cz (R.S.); 2Department of Oncology, Third Faculty of Medicine, Charles University, 128 44 Prague, Czech Republic; 3Department of Microbiology, Nutrition and Dietetics, Faculty of Agrobiology, Food and Natural Resources, Czech University of Life Sciences Prague, 165 00 Prague, Czech Republic; modrackova@af.czu.cz (N.M.); bunesova@af.czu.cz (V.N.-B.)

**Keywords:** microbiome, head and neck, HNSCC, radiotherapy, chemotherapy, immunotherapy, tumor microenvironment

## Abstract

The human microbiome plays a crucial role in the development and treatment of head and neck squamous cell carcinoma (HNSCC). Recently, a strong association between specific microbial alterations, such as increased levels of *Fusobacterium nucleatum* and *Porphyromonas gingivalis*, and the incidence of HNSCC was described. Our narrative review aims to explain how the oral microbiome influences the efficacy of various treatments, including chemotherapy and radiotherapy, noting that specific bacterial profiles may predict treatment responses. Additionally, the potential of microbiome modulation, through probiotics, to improve patient outcomes and reduce treatment-related toxicities like oral mucositis was described. Despite the promising links between microbiome composition and cancer treatment, further research to establish definitive causal relationships and therapeutic applications in HNSCC management is needed.

## 1. Introduction

Despite decades of research, cancer remains a major global health problem. In recent years, the role of microorganisms in the initiation and development of cancer has been increasingly investigated. Head and neck squamous cell carcinomas (HNSCC) currently account for ca. 4% of all malignancies, making it the sixth most common cancer worldwide. The incidence of HNSCC has been on the rise worldwide [1]. HNSCC covers a group of heterogeneous cancers of distinguished anatomical locations (oral cavity, oropharynx, pharynx, and larynx), pathoetiology, and molecular characteristics [2].

HNSCC has been associated with the carriage of several oral bacteria (e.g., *Porphyromonas gingivalis*, *Fusobacterium nucleatum*, *Streptococcus* spp.), some viruses (e.g., HPV, HHV-8, HSV, or EBV), and yeasts (*Candida albicans*). The specificity of the oral microbiome lies in the fact that a significant correlation has been demonstrated between oral dysbiosis, which can be caused by chronic alcoholism, smoking, malhygiene, microbial infections, and tumors of the oral cavity, as well as tumors of distant sites such as the oropharynx, esophagus, pancreas, colorectum, breast, prostate, or lungs [3,4,5,6].

Meta-analyses have shown that dysbiosis significantly increases the risk of oral cancer [7,8]. Among the identified microbial agents whose abundance is significantly higher in oral SCC tissue, or in other areas of the oral cavity, are, e.g., *F. nucleatum*, *Pseudomonas aeruginosa*, and *P. gingivalis* [9,10,11].

Treatment of HNSCC is complex and should be decided within the multidisciplinary team (otorhinolaryngologist, clinical oncologist, radiation oncologist, etc.). For the localized and locally advanced disease, surgery and/or (chemo)radiotherapy are the methods of choice. For recurrent and/or metastatic disease (R/M HNSCC), systemic therapy is implemented, often with the palliative goal [12].

Although little is known about oral microbial changes after surgery for HNSCC, there is information regarding gut microbiome alterations in patients after surgical treatment for colorectal cancer. A unique microenvironment resembling the microbiome of neither healthy controls nor the colorectal patients was reported [13]. Large variations exist among patients in tumor radio-responsiveness and in the incidence and severity of radiotherapy-induced side effects like dermatitis or mucositis. Exposure to ionizing radiation leads to changes in the microbiome that can contribute to more severe side effects [14,15,16,17]. The shift in microbial representation has been shown to affect the treatment response to radiotherapy and could potentially be associated with early recurrence and worse prognosis. Some studies have already proposed that the microbiome can be modified to maximize treatment response and minimize adverse effects using probiotics or prebiotics in head and neck cancer [18]. However, the oral microbiome, in terms of predictive and prognostic biomarkers, has yet to be investigated in detail.

Growing evidence supports not only the role of the microbiome in the development and progression of cancer but also the hypothesis that the host microbiome can affect anti-cancer chemotherapy efficacy and toxicity [19]. Chemotherapy (CHT) was for a long time the cornerstone of R/M HNSCC treatment; however, with the arrival of immunotherapy (immune-checkpoint inhibitors, ICIs), the treatment was revolutionized, and the therapeutic paradigm has shifted toward a preference for ICIs. Because of their structural and molecular heterogeneity, it has been challenging to develop targeted therapeutics. Cetuximab (monoclonal antibody against epidermal growth factor receptor, EGFR) remains the only targeted treatment for HNSCC [12].

Despite advances in our knowledge of the risk factors associated with HNSCC, survival rates for these cancers have not improved significantly. This only highlights the urgency of finding new means of early primary and/or tertiary prevention of these diseases. Understanding the oral microbiome and identifying risk agents; e.g., identifying protective organisms for prophylactic use or associated pathogens for targeted eradication may not only have an impact on primary prevention but could also open new possibilities for adjuvant treatment in terms of probiotic administration to reduce the risk of recurrence or metachronous cancer, or to reduce the incidence and severity of radiation-induced oral mucositis (RIOM). Ultimately, fully comprehending how the human microbiome, the oral as well as the gut microbiome, affects the response to systemic treatment in cancer patients will open new possibilities of personalized medicine enabling us to choose the most effective treatment strategy, and/or to modify a patient’s microbiome to receive the best treatment efficacy.

## 2. Methods

A literature search was conducted in PubMed/Medline (prior to 29 June 2024) to retrieve relevant articles on this topic. The keywords used for database search were “microbiome”, “head and neck cancer”, “head and neck squamous cell carcinoma”, “radiotherapy”, “chemotherapy”, “cetuximab”, “immunotherapy”, and “surgery”. Only English language articles were considered. Due to the nature of this narrative review, we prioritized the most recent studies. Regarding the type of studies, umbrella reviews, meta-analyses, and systematic reviews were prioritized, in that order, over individual studies. We excluded letters to the editor, editorials, and conference reports.

## 3. Results

### 3.1. Microbial Risk Factors of Developing Head and Neck Cancer

Although viral oncogenesis has been extensively studied since the 20th century, scientific knowledge of bacterial oncogenesis has not been well elucidated [20]. The first reports about oncogenic microbiome date back to the 19th century, when Virchow described pro-cancerogenic effects of Schistosoma on the urinary bladder [21]. His hypothesis of chronic inflammation remains one of the widely supported mechanisms of bacterial carcinogenesis, the other ones being antigen-driven proliferation and metabolite secretion (hormones, oncoproteins, and/or toxic metabolites) [20,22].

Emerging evidence promotes oral microbes as potential biomarkers for various malignancies, e.g., of the oral cavity as well as of distant sites, and even premalignant lesions [23]. The most common premalignant lesions in the oral cavity are leukoplakia, erythroplakia, and erythroleukoplakia, sometimes labeled as oral potentially malignant disorders (OPMD) [24]. Identifying specific oral microbiota associated with OPMD and/or early-stage HNSCC could initiate research efforts to develop screening programs. Such an initiative could reduce morbidity and mortality as the proper treatment could be started in earlier stages. Lee et al. reported significant differences in the microbiome of healthy controls, patients with OPMD, and patients with oral squamous cell carcinoma (OSCC). They demonstrated the significantly higher abundance of *Fusobacterium* spp., *Prevotella* spp., *Porphyromonas* spp., *Veillonella* spp., *Actinomyces* spp., *Clostridium* spp., *Haemophilus* spp., *Streptococcus* spp., and *Enterobacteriaceae* in patients with OPMD and OSCC compared to the healthy controls. Furthermore, they reported significant differences in the abundance of *Bacillus* spp., *Enterococcus* spp., *Parvimonas* spp., *Peptostreptococcus* spp., and *Slackia* spp. between OPMD and OSCC [25]. This can imply that the microbiome plays a crucial role in the carcinogenesis of OSCC. If we could identify the causative mechanism behind OPMD development or its transformation into OSCC, we could design a proper intervention and thus modify the oral microbiome to reduce the chances of HNSCC development [26,27,28,29].

HNSCC represents a heterogeneous group of diseases of various anatomical sites, cellular origin, and different pathoetiology [30]. Poor oral health and oral dysbiosis are associated with tumors of the oral cavity, as well as tumors of distant sites [3,4,5,6]. Such dysbiosis leads to immune dysregulation, which contributes to the development of various diseases, including systemic infections and solid and hematological malignancies [31,32]. The oncogenic microbiome has synergistic effects with other carcinogens. Several studies have described that certain oral microbiome alterations caused by smoking, hygiene, diet, and alcohol consumption enhance tumor development; however, no consensus data exist at this time [7,8,24,33,34,35,36,37]. Frank et al. reported a significantly higher diversity of oral microbiome in HNSCC compared to healthy controls; moreover, in HNSCC patients, significant interindividual variability was observed across bacterial profiles [38]. Some concordance has been found in the varied data; however, no clear pattern has yet been revealed. The reasons behind such high variety in results can be many, e.g., sample type (oral swabs, saliva, tumor tissue samples, etc.), tumor stage, comorbidities, variations in gut microbiome, etc. Among the identified microbial agents significantly associated with the development of oral cancer are *P. gingivalis*, *F. nucleatum*, and *Ps. aeruginosa* [9,10,11]. Wang et al. observed significant differences in the abundance of *Actinomyces* spp. and *Parvimonas* spp. in HNSCC patients and healthy controls. In the HNSCC patients’ microbiome, Actinomyces spp. were rather depleted, and the representation of *Parvimonas* spp. was higher compared to the healthy controls. Moreover, these results correlated with the T stage of tumors [39].

As it seems now, many distinctive pro-cancerous mechanisms are applied in microbial oncogenesis, ultimately leading to a proinflammatory state, e.g., production of lipopolysaccharides, enzymes, bacterial toxins, etc. [40]. *Streptococcus salivarius*, *Corynebacterium* spp., and *Stomatococcus* spp. show strong oxidizing properties, which they utilize to convert alcohol into acetaldehyde, which can then interfere with DNA synthesis and repair mechanisms [41,42]. *P. gingivalis* stimulates the production of dendritic precursor cells, which inhibit cytotoxic CD8+ lymphocytes; it also stimulates the transcription of proteins facilitating cell division, subsequently promoting cell proliferation [43,44].

Incongruent with the above-mentioned studies, Kumpitsch et al. reported no significant difference in the diversity of microbiota between HNSCC patients and controls; however, significant differences in the abundance of microbes were reported—the abundance of the genera *Veillonella*, *Rothia*, and *Haemophilus* from the phylum Firmicutes was significantly higher in HNSCC patients [45]. Similarly, Mäkinen et al. observed a direct association of OSCC and the oral microbiome. Microbial diversity was not significantly affected, but the relative abundance of *S. anginosus*, *Abiotrophia defectiva*, and *F. nucleatum* was significantly higher in HNSCC patients. On the contrary, *Prevotella histolitica*, *Haemophilus parainfluenzae*, and *F. periodonticum* were dominantly represented in the healthy cohort [46]. Huang et al. observed a specific microbiome in the HNSCC tumor tissue. On the surface of oral SCCs, both anaerobes, e.g., *Actinomyces* spp., *Clostridium* spp., *Fusobacterium* spp., *Porphyromonas* spp., and *Bacteroides* spp., and aerobes, e.g., *Klebsiella* spp., *Citrobacter* spp., *Streptococcus* spp., *Enterobacter* spp., and *Serratia* spp., were identified. *S. anginosus* colonizes dental plaques and produces nitric oxide and cyclooxygenase-2, facilitating DNA damage [47].

Across most of the studies, a similar pattern of increased abundance of *Lactobacillus* spp. and decreased abundance of *Neisseria* spp. was observed in HNSCC patients compared to healthy controls [48,49,50,51,52,53,54,55]. These two genera are consistently found differentially abundant in oral, oropharyngeal, and laryngeal cancers, suggesting potential cancer-protective properties of *Neisseria* spp. and cancer-promoting properties of *Lactobacillus* spp. Frank et al. established an OSCC model in mice by chemically induced carcinogenesis (4-nitroquinoline-1-oxide, 4NQO), in which a direct link between oral dysbiosis and OSCC dynamics was demonstrated. As they proposed, a myriad of mechanisms are involved in this bacterial-induced cancerogenesis. They hypothesized that *Lactobacillus* spp. promotes cancer development via production of tryptophan and its metabolites, which act as a cancer-promoting aryl-hydrocarbon receptor (AhR) pathway activator. AhR activation is a critical pathway in many cancers, including HNSCC, which enhances cancer stem cell migration [38].

Healthy oral microbiome composition is also crucial for preventing periodontitis, which is strongly associated with HNSCC development [56]. Dental plaque biofilms cause chronic inflammation, which leads to the irreversible destruction of tooth-supporting tissues. This process is facilitated via many bacteria; however, *F. nucleatum* appears to play the crucial role. *F. nucleatum* is essential in establishing polymicrobial plaques. It helps to navigate other bacteria, such as streptococcal species, *S. anginosus*, or *P. gingivalis*, into the plaque [57]. Currently, only one meta-analysis investigated the plausible association between *F. nucleatum* and HNSCC. Bronzatoa et al. reported its higher abundance in HNSCC patients compared to controls. However, as the authors discussed, its presence can be only a result of a hypoxic environment within the dental plaque, and no direct evidence supporting the hypothesis of *F. nucleatum*-mediated tumor transformation exists [58]. On the other hand, *F. nucleatum* directly interacts with epithelial cells, promoting cancerogenesis, faster growth, and spread of HNSCC. *F. nucleatum* also produces FadA adhesins, which bind to E-cadherin receptors on endothelial cells, enabling them to hematogenously transfer to distant niches [59].

The prognostic effects of *F. nucleatum* on disease/relapse-free survival in HNSCC were investigated by two independent trials. They reported that *F. nucleatum* was significantly associated with longer overall survival and a lower relapse rate [60,61]. These findings are quite unexpected, especially considering its association with higher aggressiveness and worse prognosis in other cancer types, e.g., esophageal, colorectal, and pancreatic carcinomas [62,63,64,65,66]. The reason why is currently unknown; however, *F. nucleatum* was observed to be significantly more abundant in HPV+ HNSCC, which represents a distinctive entity. HPV+ HNSCCs usually affect younger people, non-smokers, non- or just slight alcohol consumers, and are associated with better prognosis [60,61]. Data on bacterial oncogenesis in these patients are limited and require further scientific research.

### 3.2. Pathologic Mechanisms of F. nucleatum-Facilitated Impact on HNSCC

The impact of *F. nucleatum* on HNSCC development has recently become an object of scientific interest and research. Carcinogenic properties of *F. nucleatum* have mostly been attributed to its interaction with Toll-like receptors (TLRs) of oral epithelium and to its pro-inflammatory effects and are depicted in Figure 1 [67]. Lipopolysaccharides of the cell membrane of *F. nucleatum* can directly bind to the TLR4 of epithelial cells and thus activate the NF-κB signaling pathway, leading to the over-expression of various cytokines such as tumor necrosis factor-alpha (TNF-α), interleukin-6 (IL-6), IL-8, IL-10, IL-12, and reactive oxygen species (ROS) production, similar to that observed in colonic cancer [67,68,69,70]. OSCC infiltrated with *F. nucleatum* have also been observed to exhibit increased invasiveness, migration, and proliferation of cancer cells [71,72,73]. *F. nucleatum* was observed to facilitate functional loss of E-cadherin (CDH1), increased cell migration, and the upregulation of Snail family transcriptional repressor 1 (SNAI1) in cancerous as well as non-cancerous oral epithelial cells, indicating epithelial–mesenchymal transition (EMT) [74].

How and when *F. nucleatum* colonizes HNSCC tissue is still unclear. Recent evidence supports that *F. nucleatum* uses its Fap2 adhesin to bind to a Gal-GalNAc oligosaccharide on cancer cells; however, these findings come from breast and colon cancer models [75,76]. No studies have yet investigated whether OSCC colonization with *F. nucleatum* is facilitated via the Fap2/Gal-GalNAc mechanism, although indirect evidence of Gal-GalNAc overexpression in OSCC cells coinciding with increased abundance of *F. nucleatum* supports this theory [77,78].

The CDH1 pathway seems to play a pivotal role in accelerating the OSCC cell cycle and stimulating cell proliferation [79]. In preclinical models, higher representation of phosphorylated CDH1 (pCDH1) was found in OSCC cells infiltrated with *F. nucleatum*. pCDH1 results in the upregulation of β-catenin, Myc, and Cyclin D1, subsequently inducing cell proliferation. The pathway is summarized in Figure 2. Cyklin D1 is often overexpressed in OSCC and is associated with tumor progression, higher grade, and poor prognosis [79,80,81]. Interestingly, no effect of *F. nucleatum* on CDH1/β-catenin in noncancerous cells was observed, which suggests that *F. nucleatum* is not responsible for cancerogenic transformation of the oral epithelium [79]. This discovery could imply that *F. nucleatum* is not responsible for the cancerogenesis of OSCC, even though a higher representation of *F. nucleatum* was observed in OSCC samples compared to healthy individuals [58]. However, enriched representation of *F. nucleatum* was also observed in OPMDs, which proves that colonization of OSCC/HNSCC with *F. nucleatum* begins much earlier in the process of cancerogenesis and potentially supports the role of *F. nucleatum* in this process [77,78].

*F. nucleatum* has been observed to play a crucial role in enhancing the invasiveness of OSCC. Via direct interaction of Fap2 with epithelial cells‘ TLRs, the IL-6-STAT3 signal pathway is augmented, leading to the upregulation of matrix metalloproteinases such as MMP1 and MMP9 [67]. Overexpression of MMPs leads to excessive degradation of the extracellular matrix, which facilitates cancer cells’ invasiveness and dissemination [82,83,84,85].

Moreover, *F. nucleatum* has a unique capability of suppressing the immune system in the tumor microenvironment (TME) via the inhibition of immune cells’ activity [86]. An inverse association between *F. nucleatum* representation and abundance of CD3+ and CD4+ T-lymphocytes was observed in breast and colon tissue. Fap2 adhesin binds to the T-cell immunoreceptor with Ig and ITIM domain (TIGIT), which is an important immunomodulatory receptor of T-cells and NK-cells, and inhibits it [87]. This Fap2-TIGIT interaction protects *F. nucleatum* from the immune system response, and furthermore, it protects the surrounding cancer cells via a bystander effect [67,87,88,89,90]. Fap2 is also linked to the cell death of lymphocytes, further promoting its local immunosuppressive effects [90].

Fap2 adhesin is one of the most significant virulence factors of *F. nucleatum*. As depicted in Figure 3, it facilitates the colonization of cancer tissues, promotes its invasiveness, migration, and proliferation, and suppresses the local immune response, but it also enables the *F. nucleatum* to form biofilms through aggregation of other bacterial species [91]. Certain bacteria, e.g., *P. gingivalis*, are not able to survive in the OCSS environment alone. *F. nucleatum* can produce ammonia to neutralize the local pH, which enables acid-sensitive bacteria to co-exist in *F. nucleatum* biofilm. Whether such co-aggregations have a direct impact on cancerogenesis is unclear, as studies report incongruent results. Binder Gallimindi’s group observed significantly higher proliferation of OSCC cells co-cultivated with the mixture of *P. gingivalis* and *F. nucleatum*; however, Harrandah and colleagues reported that OSCC cells infected with *F. nucleatum* alone exhibited a higher proliferation rate compared to the combination of periodontal pathogens [67,92].

### 3.3. Microbiome and Radical Treatment—Surgery and (Chemo)Radiotherapy

Treatment of localized and locally advanced HNSCCs is complex and should be debated within a multidisciplinary board. Treatment modality is chosen based on the stage, localization, and histopathological characteristics of the tumor. In early stages, surgery alone can be sufficient, and in more aggressive or extensive cancer, adjuvant radiotherapy (RT) is applied. Sometimes, radical chemoradiotherapy (CRT) can offer comparable results with surgery and adjuvant treatment, however, saving patients from possibly mutilating surgery. Each of these modalities can impact the oral microbiome [12].

A very limited number of studies linked to the perioperative changes in the oral microbiome have been published. One study described a higher abundance of *Haemophilus* spp., *Neisseria* spp., *Aggregatibacter* spp., and *Leptotrichia* spp. in the post-resection microbiome of HNSCC patients [49]. In case of successful resection, these bacterial alterations reverted to normal within three months since the curative surgery [93].

RT of HNSCC is associated with a high incidence of RT-induced toxicity, acute as well as late, e.g., xerostomia, radiation-induced oral mucositis (RIOM), dermatitis, dysphagia, etc., which can significantly affect patients’ Quality of Life (QoL) [94,95,96]. The intensity and/or severity of these adverse events is enhanced with CRT [97]. It remains unclear whether oral dysbiosis is associated with (C)RT-related complications and/or treatment efficiency. In this chapter, we will discuss the impact of the oral microbiome on (C)RT efficacy and acute toxicity. Its role in the genesis of late toxicity (neuromuscular damage, fibrogenesis, etc.) has not been published so far [98].

The most common RT-induced adverse event in HNSCC patients is xerostomia. Xerostomia is caused by high radiation doses to salivary glands, which leads to significantly decreased salivary flow [97]. Xerostomia thus often leads to impaired speech and taste, and malnutrition, overall diminishing the QoL of HNSCC patients. Salivary flow is insufficient in delivering alkaline salivary solutions, thus lowering intraoral pH, which predisposes HNSCC patients to develop oral mucositis, dental caries, mucosal ulcerations, and oral infections [99]. Interrupted salivary flow leads to microbial alterations in the oral cavity, resulting in higher relative abundances of *S. mutans*, *Lactobacillus* spp., *Candida* spp., and *Staphylococcus* spp.; in contrast, the relative abundance of *S. sanguinis*, *Neisseria* spp., and *Fusobacterium* spp. decreases [74]. Mojdami et al. later reported results in agreement with previous studies. They observed an increased relative abundance of the genera *Streptococcus*, *Lactobacillus*, *Treponema*, and *Prevotella* [14]. Similarly, Chen et al. observed a significant reduction in the relative abundance of *Haemophilus* spp., *Veillonella* spp., and *Granulicatella* spp., and an increase in the relative abundance of genera *Lactobacillus*, *Scardovia*, *Acinetobacter*, and *Enterococcus* after CRT [45].

RIOM represents a common debilitating toxicity associated with RT/CRT of the head and neck region, affecting ca. 80–90% patients, among whom up to 60% may develop severe RIOM (Grade 3 or 4) [100,101,102]. In recent years, growing evidence has shown that oral dysbiosis contributes to the incidence and severity of RIOM [16,103,104,105]. Hu et al. demonstrated a dose-dependent oral microbiome alteration after RT. Higher administered doses were associated with an increased abundance of *Pseudomonas* spp., *Treponema* spp., and *Granulicatella* spp., and a decreased abundance of *Prevotella* spp., *Fusobacterium* spp., *Leptotrichia* spp., *Campylobacter* spp., *Peptostreptococcus* spp., and *Atopobium* spp. Pre-existing dysbiosis can predispose patients to developing more severe RIOM. Patients who during RT developed RIOM Gr 2 and higher had an increased abundance of several Gram-negative bacteria (e.g., *Fusobacterium* spp., *Haemophilus* spp., *Tannerella* spp., *Porphyromonas* spp., and *Eikenella* spp.) before RT [15]. Such microbial alterations subsequently lead to microbial imbalance, which results in deterred immunomodulatory functions and promotes chronic inflammation. CRT-induced cell apoptosis results in oral mucosal damage, which is then exacerbated by the amplified inflammatory cascade [102,105].

Regardless of the acute toxicity, certain microbial alterations after RT can be observed in all HNSCC patients. Gaetti-Jardim et al. reported a reduced abundance of Gram-negative obligate anaerobes and an enhanced abundance of *S. mutans* [106]. Studies investigating microbial changes between the pre- and post-RT microbiome concordantly described a dose-dependent reduction in bacterial diversity after RT. The reconstitution of microbial diversity to the pre-RT composition has been observed with the prolonging time since the end of RT [107].

RT for HNSCC affects oral cavity epithelia directly through ionizing radiation and indirectly through reduced salivary flow, both contributing to cariogenesis [11,14,16,17]. RT-induced dental issues can significantly deteriorate the QoL of HNSCC patients and can lead to osteoradionecrosis of the jaw. Regular dental follow-ups are necessary, and specialized dental care should be provided to all HNSCC patients who underwent RT/CRT [108]. Vesty et al. suggested oral probiotics containing *Streptococcus salivarius* M18 to modulate host immune response and inter-bacterial cross-talk to improve oral health; unfortunately, they failed [109].

So far, five studies have been conducted on probiotics and prebiotics administration to prevent or alleviate symptoms of RIOM [110,111,112,113,114]. Probiotics can competitively inhibit pathogens by the production of bacteriocins, and they can also modulate cell proliferation and apoptosis. Prebiotics stimulate the growth of beneficial microbiota and can also inhibit the activity of some bacterial pathogens. The probiotic strain *Bifidobacterium animalis* subsp. *lactis* HN019 has been shown to play a role in modulating the immuno-inflammatory response of the human body, benefiting oral health, alleviating the symptoms of periodontitis, and shortening the duration of its treatment [115]. In a recent study, the administration of probiotics based on *Bifidobacterium* spp. and *Lactobacillus* spp. was shown to significantly reduce the grade of radiotherapy-induced mucositis (RIOM) [116]. Also, the exopolysaccharide produced by *Bifidobacterium breve* was shown to have antitumor effects against HNSCC, which could lead to the future inclusion of probiotics with *B. breve* in the adjuvant therapy of HNSCC [116,117]. Sharma et al. observed a significant decrease in severe RIOM incidence in the probiotics group (administered *Lactobacillus brevis* CD2) compared to the placebo group (52 vs. 77%) [111]. Limaye et al. observed that the group of HNSCC patients, to whom the prebiotic *Lactococcus lactis* was given, had a 35% reduction in ulcerative RIOM prevalence throughout the RT (15.5 vs. 45.7%) [112].

As oral dysbiosis can significantly affect the morbidity of HNSCC patients undergoing RT, some authors were interested in whether pretherapeutic antibiotics administration could affect the incidence of RIOM and/or treatment results [118]. RT-induced mucositis is associated with the local microbiome. It was demonstrated that germ-free mice exhibited a higher radioresistance of their intestinal mucosa compared to regular mice, whose radiotherapy-related toxicity was much higher [111,119]. One single-center analysis reported peri-therapeutic antibiotics administration as an independent prognostic factor of reduced disease-free survival [120]. Just recently, Rühle et al. published an analysis on 220 HNSCC cancer patients undergoing definitive CRT. They reported that patients pretreated or treated concurrently with RT with antibiotics had significantly lower disease-free and even overall survival (OS)—the median of OS was 10 months shorter (36 vs. 26 months) [118]. Altogether with the Nenclares group, data on 492 HNSCC patients agreeably demonstrated a worse prognosis of HNSCC patients (pre)-treated with antibiotics. We could discuss how the overall survival could be lower because of the co-morbidities requiring the antibiotics therapy (e.g., pneumonia, port infection, etc.); however, in both cohorts, an independent significant association was found between antibiotics use and decreased progression-free survival [118,120].

Alterations of bacterial representation in the oral microbiome have been shown to affect the treatment response to RT. *F. nucleatum* has been suggested as a predictor of the therapeutic response of HNSCC to RT [121]. Mechanisms underlying its impact have not been elucidated; however, the ability of *F. nucleatum* to aggregate and to form biofilms has been speculated to increase radioresistance by attracting neutrophils and cytokine release [122,123].

Plausible microbiome modifications to maximize the treatment response and minimize adverse effects with probiotics in HNSCC have already been proposed; however, the proper identification and characterization of protective and/or prognostic factors have yet to be investigated in detail [18,124,125].

### 3.4. Microbiome and Systemic Therapy for HNSCC

Treatment of HNSCC requires a multimodal approach. For the localized and locally advanced stage of HNSCC, either definitive CRT, especially for locally advanced disease (LA), or primary surgical approach followed by an adjuvant RT is the standard of care [12]. However, despite the technological progress that RT has made in the last 20 years (IMRT, IGRT, VMAT techniques), the recurrence rates are still quite high. Recurrence rates for primary locally advanced HNSCC can vary depending on many factors, including the specific tumor location, extent of disease, response to treatment, and other clinical factors. Nevertheless, the following overview can be provided [126].

Patients with locally advanced HNSCC treated with radical CRT have a recurrence rate of around 30–50%. Outcomes may vary depending on the exact treatment protocol and the individual patient’s response to therapy [127]. Patients treated with surgery followed by adjuvant RT (with or without chemotherapy) have a recurrence rate of approximately 30–40%. Again, outcomes may vary depending on the specific treatment conditions and other factors [128]. Patients at high risk of recurrence, i.e., especially patients with observed lymph node extracapsular extension, may benefit from the addition of concurrent CHT to adjuvant RT [129].

It is important to emphasize that individual prognosis and recurrence rates may vary considerably depending on the specific clinical characteristics of each patient. The multidisciplinary team takes many factors into account, especially the performance status of the patients, comorbidities—renal insufficiency, chronic cardiac failure, autoimmune diseases, etc., molecular characteristics of the tumor, e.g., expression of programmed-death ligand 1 (PD-L1) either on tumor cells (tumor positive score, TPS) or on both tumor and immune cells in the tumor (combine positive score, CPS), or next-generation sequencing (NGS) results, when deciding the most appropriate treatment approach and assessing the risk of recurrence. As for R/M HNSCC, this stage is usually incurable. Currently, for R/M HNSCC, the most used regimens include monotherapy or a combination of chemotherapeutics such as cisplatin (cDDP), carboplatin (CBDCA), 5-fluorouracil (5-FU), docetaxel, paclitaxel, immune-checkpoint inhibitors as pembrolizumab and nivolumab, and targeted treatment agents such as cetuximab (anti-epidermal growth factor receptor monoclonal antibody). Currently, the immunotherapy strategies are preferred to the targeted therapy in HNSCC [12].

#### 3.4.1. Chemotherapy

Among the most often used chemotherapeutics in the treatment of R/M HNSCC belong alkylating agents (cDDP, CBDCA), nucleic acid inhibitors (5-FU), and microtubule depolymerization inhibitors (docetaxel and paclitaxel) [130,131,132,133,134,135,136].

The human microbiome plays a significant role in the development and progression of cancer; moreover, it was proven that the host microbiome can also affect anti-cancer CHT efficacy and toxicity. Alexander et al. proposed three possible outcomes of microbial impact on chemotherapeutics: direct influence on drug efficacy through pharmacokinetic alterations; abrogation and compromise of anticancer effects; and mediation of toxicity [19].

Animal studies have shown that manipulation of the microbiome can affect the efficacy and toxicity of CHT [137,138]. In animal models, cisplatin efficacy and toxicity severity and/or incidence have been found to be associated with specific microbial alterations [137,139,140]. In the Lewis lung carcinoma mouse model, Gui et al. observed significant survival reduction in mice treated with cisplatin and concurrent antibiotics (vancomycin, ampicillin, and neomycin) compared to cisplatin alone. In contrast, mice that were given cisplatin concurrently with *Lactobacillus* spp. exhibited better a cisplatin response rate, more rapid shrinkage of the tumor, and longer survival rate compared to the mice treated with cisplatin alone [141]. Chen et al. similarly investigated the effect of different microbial profiles on cisplatin efficacy in the Lewis lung carcinoma mouse model, reporting that *Akkermansia muciniphila* enhances the anti-cancer effects of cDDP. Compared to cDDP alone, combined treatment with *A. muciniphila* led to slowed-down tumor growth, downregulation of Ki67 and p53, and upregulation of pro-apoptotic proteins [142]. Hsiao et al. observed attenuated nephrotoxicity of cisplatin in Wistar rats, which were given cisplatin concurrently with *Limosilactobacillus reuteri* and *Clostridium butyricum* [143]. cDPP is usually preferred to CBDCA in HNSCC treatment, and it does not appear to be an attractive target for (pre-)clinical microbial research. However, one Chinese study described higher intestinal toxicity in mice treated with CBDCA, which had a higher abundance of *Prevotella capri* in their gut microbiome [144].

5-FU is usually used in combination with platinum-based CHT, usually cDDP, in the first or second line of R/M HNSCC [12]. Yuan et al. performed similar experiments to Gui and collaborators, as described above. They investigated the treatment efficacy of 5-FU in a mouse model. Mice treated concurrently with 5-FU and a combination of antibiotics (vancomycin, ampicillin, and neomycin) had a much worse treatment response compared to mice treated only with 5-FU; the use of probiotics had no improvement in the response rate [145]. Yu et al. investigated the impact of the gut microbiome on CHT efficacy in colorectal cancer. They reported that *F. nucleatum* increases chemoresistance to 5-FU in colorectal cancer by modulating autophagy [146]. Certain probiotics exhibited protective properties against intestinal mucositis, suggesting their potential use in the alleviation of toxicity. Use of the strain *Limosilactobacillus fermentum* BR11, probiotic product VSL#3 (containing *S. thermophilus*, *B. breve*, *B. longum* subsp. *longum* and *B. longum* subsp. *infantis*, *L. delbrueckii* subsp. *bulgaricus*, *L. acidophilus*, *Lacticaseibacillus paracasei*, and *Lactiplantibacillus plantarum*), and species *Lacticas. casei* and *Lacticas. rhamnosus* are associated with reduced chemotoxicity, reduced diarrhea, and weight loss in rats [147,148,149,150]. Similar effects were observed with *L. acidophilus*, *Lacticas. rhamnosus*, and *Lacticas. paracasei*, and with *B. animalis* subsp. *lactis*, *B. bifidum*, and *B. longum* subsp. *infantis* when used concurrently with 5-FU [151,152,153].

Clinical studies on the relationship between CHT and the host microbiome in head–neck cancer are either lacking or focus on CRT. However, one small Chinese study on 44 OSCC patients investigated how the microbiota affects the efficacy of induction CHT with TPF (docetaxel, cisplatin, and 5-fluorouracil). Rui et al. observed different microbial profiles in responders and non-responders. In the responder group, *Slackia* spp. abundance was enriched; among the non-responders, *Mycoplasma* spp. and *F. nucleatum* were enriched [154]. In accordance with the study by Yu and collaborators, these findings suggest that *F. nucleatum* could promote chemoresistance even in head and neck cancers [144].

The microbiome plays an important role in modulating the effects of CHT through modulation of the immune system, drug metabolism, intestinal barrier protection, and the production of toxic metabolites [155,156,157]. Limited data are available on the impact of CHT on the oral microbiome of HNSCC patients. CHT alone affects the oral microbiome, decreases the abundance of *Streptococcus* spp., and causes a relative increase in the abundance of Gram-negative anaerobic bacteria [158]. It can also lead to dysbiosis, resulting in metabolic changes, e.g., dysregulation of the enterosalivary nitrate-nitrite-nitric oxide pathway, leading to biochemical changes in the mouth cavity. Suggesting that these changes may alter HNSCC patients’ risk of recurrence [159]. Manipulation of the microbiome represents a potential therapeutic strategy to improve the efficacy of CHT and reduce its side effects.

A literary review regarding the influence of the gut microbiome on systemic anti-cancer therapy is summarized in Appendix A.

#### 3.4.2. Targeted Therapy

Cetuximab is a monoclonal antibody targeting epidermal growth factor receptor (EGFR) and is approved for HNSCC treatment [160]. As far as authors are concerned, no studies investigating how the microbiome alters the toxicity or treatment results of cetuximab monotherapy have been published. The lack of evidence might be caused by the declining usage of cetuximab in HNSCC treatment. CRT with cDDP has been found superior in terms of treatment response, as well as tolerability; moreover, immune checkpoint inhibitors also surpassed cetuximab-based regimens in palliative settings [161]. The efficacy of targeted treatment after ICIs is being investigated by a phase II trial (NCT04375384) [162].

Only two single-arm phase II clinical trials, CAVE-mCRC and CAVE-LUNG, investigated cetuximab in combination with ICI avelumab in metastatic colorectal carcinoma (mCRC) and chemo-refractory non-small cell lung carcinoma (NSCLC) patients, respectively. Specific taxa, *Agathobacter rectalis* M104/1 and *Blautia* sp. SR1/5, were identified in long-term responding patients in both trials [163], as to whether these microbiotas can affect the treatment response in HNSCCs requires further evaluation.

#### 3.4.3. Immunotherapy

Since the landmark trials, KEYNOTE-048 and CheckMate141, immunotherapy with ICIs has become the cornerstone of R/M HNSCC palliative treatment in the first and second line [12,164,165]. Currently, two ICIs are routinely used in HNSCC treatment, i.e., nivolumab and pembrolizumab. Both IgG4 monoclonal antibodies target programmed death receptor-1 (PD-1) on the cytotoxic CD8+ T-lymphocytes, diminishing inhibitory signals from the programmed death ligand-1 (PD-L1) exhibited on the cancer cell surface [166,167]. Therefore, CD8+ T-lymphocytes can be subsequently activated and thus eliminate cancer cells [168]. Even though immunotherapy revolutionized treatment across most of the solid tumors, its objective response rate is usually around 40% [169,170]. Many studies have already demonstrated how closely the human microbiome and immunotherapy are linked, suggesting it could be connected to the primary and/or secondary (acquired) resistance to the ICIs [171,172,173,174,175,176,177,178,179]. Common effort is currently aimed at identifying “cold” and “hot” microbial constitutions that could predict the ICI treatment response and could eventually lead to attempts on how to modify the human microbiome to enhance treatment efficacy. Some studies have already tried to manipulate the gut microbiome for therapeutic purposes through, e.g., fecal transplantation, probiotics use, antibiotics administration, etc. [162].

The association of ICI efficacy and microbial compositions stems from pre-clinical research on a mouse model. Vetizou et al. reported germ-free mice as non-responders to ICIs. Once these mice received either oral gavage or fecal transplantation containing *Bacteroides fragilis*, they became ICI-responsive [180]. Similar results were reported in other studies, regardless of the target of ICIs (PD-(L)1 or CTLA-4, i.e., cytotoxic T-lymphocyte-associated protein [171,181,182]. Most current clinical studies investigated the effect of gut microbiome on ICI efficacy; however, their results were, for a long time, incongruent. Broad-spectrum antibiotics were shown to disrupt gut microbiota and thus reduce the efficacy of ICIs; however, non-targeted commercial prebiotics also did not improve ICI efficacy [183,184,185]. Also, the exact mechanism of how the gut microbiome influences ICI efficacy is still unknown.

Currently, the most attention regarding the impact of the gut microbiome on ICI treatment efficacy is paid to metastatic melanoma, especially to enhance the rate and depth of the response [162,183,186]. Only limited data are available on the role of the gut microbiome and its metabolome on ICI efficacy in R/M HNSCC. Bari et al. reported significantly higher bacterial diversity in their gut microbiome, a lower Firmicutes-to-Bacteroidetes phylum ratio, and a higher abundance of the genus *Bacteroides* and family *Lachnospiraceae*; uniformly, the responders had a higher abundance of *Eubacterium oxidoreductans* and *Bact. uniformis* in the baseline and post-treatment samples, and *Ruminococcus* spp. was enriched in the durable responders. Interestingly, they also found a different metabolomic signature. Significantly lower levels of adenosine, inosine, and xanthine were reported in baseline samples of responders; in this group, adenosine, inosine, and xanthine levels continued to decrease, and vice versa, the levels rose in non-responders throughout the therapy, suggesting that inosine is consumed by the activated T-cytotoxic cells [187]. Despite promising results from other solid tumors, a recent pilot study failed to find an association between the oral microbiome and nivolumab treatment response in HNSCC patients. Farris and colleagues observed no associations in bacterial diversity (baseline and week 7) and clinical response, PD-L1 expression, HPV status, and treatment duration [188].

The key issue currently is the difference and diversity in the data obtained. Many studies identified the differences in microbial composition affecting ICI efficacy; however, all the studies failed to agree on specific universal bacterial agents [171,172,173,174,175]. Moreover, even a meta-analysis of these studies failed to do so [189].

## 4. Discussion

### 4.1. Bacterial Alterations in HNSCC

Traditionally, HNSCC, especially arising in the oropharynx, can be stratified according to the evidence of HPV-infected cells, HPV-positivity being the harbinger of radiosensitivity, and of better prognosis [190]. Recently, bacterial oncogenesis has been profoundly studied; however, no convincing data have yet been reported. Several obstacles were reported: (a) HNSCC is a highly heterogeneous group of diseases that may differ in anatomical sites, its pathoetiology, and histopathology; (b) high interindividual variability is typical even for the healthy oral microbiome; and (c) often inconsistent results were reported. As we already addressed, different investigators reported either higher or unchanged bacterial diversity in HNSCC patients and healthy controls. Surprisingly, investigators often reported even contradictory findings of increased/decreased relative abundance in the HNSCC patients compared to controls [7,8,34,35,36,45,46,47]. Across many studies, an increased abundance of *F. nucleatum*, *P. gingivalis*, *Ps. aeruginosa*, *Lactobacillus* spp., and a decreased representation of *Neisseria* spp. and *Actinomyces* spp. were observed [9,10,11,25,38,45,46,47,48,49,50,51,54,55].

Microbial alterations discovered in HNSCC patients are summarized in Appendix A.

Particularly, a large amount of evidence has proved that *F. nucleatum* is closely related to tumor development [191]. *F. nucleatum* usually inhabits the oral cavity and causes periodontal disease [64,146,192]. A significant correlation has been demonstrated between oral dysbiosis, especially the increased abundance of *F. nucleatum*, and tumors of the oral cavity, as well as tumors of distant sites such as the oropharynx, esophagus, pancreas, gut, breast, prostate, or lungs [3,4,5,7]. *F. nucleatum*-associated tumors are generally linked with worse treatment response, higher aggressiveness, and worse prognosis [62,193]. Surprisingly, it appears to be the opposite in HNSCC. HNSCC tested positive for *F. nucleatum*, had a lower recurrence rate, and a prolonged overall survival [61]. The reasons why are currently unknown, although microbial inhibition might be involved.

The genus *Bifidobacterium* occurs naturally in the oral and intestinal lumen [194]. Bifidobacteria have combined local and systemic effects, including competitive inhibition of pathogenic bacteria, production of organic acids and bacteriocin-like compounds, and immunomodulation [118,195,196]. A probiotic formula containing the species *B. adolescentis*, *B. longum*, and *B. bifidum* exhibited inhibitory effects on the growth of *F. nucleatum* in vitro [197]. This was confirmed in another independent study, where a direct effect on CRT efficacy and toxicity was observed [198]. Details of how bifidobacteria interact with *F. nucleatum* remain unclear; however, one of the possible anti-cancer mechanisms includes the production of exopolysaccharides, which stimulate the Bax gene pathway, resulting in the overexpression of Caspase 3 and 8, resulting in the induction of apoptosis [109,110]. Whether cross-talk between bifidobacteria and *F. nucleatum* in the oral cavity is responsible for the better treatment response in *F. nucleatum*-associated HNSCC still lacks any evidence.

### 4.2. Human Microbiome Has a Direct Influence on Anti-Cancer Therapy in HNSCC

Recently, the human microbiome has been vigorously researched as its impact on anti-cancer treatment efficacy appears to be eminent [155]. Mounting evidence supports that specific gut microbiome composition affects treatment response to various chemotherapeutics, e.g., gemcitabine, cyclophosphamide, oxaliplatin, etc. [146,159,199,200,201,202]. In the case of HNSCC, almost all data came from animal models. The only published clinical study investigated the treatment response to the induction of TPF chemotherapy. Interestingly, *F. nucleatum* was significantly more abundant in the non-responder group [154]. Based on the literature review, *F. nucleatum* interferes with the metabolism of 5-FU and thus induces chemoresistance in CRC [146]. Hypothetically, this can be a similar case. An interesting concept of inflammation-induced chemoresistance was proposed based on a mouse model. Mice with OSCC had orally inoculated *P. gingivalis*, which resulted in chronic inflammation. These mice were then treated with paclitaxel or docetaxel, with or without anti-inflammation drugs. Mice who were infected with *P. ginigvalis* were more chemoresistant compared to the uninfected mice; however, the infected mice treated concurrently with antiinflammation medication became chemosensitive to paclitaxel/docetaxel again [203,204]. An interesting finding of the ability to alleviate toxicity and enhance CHT efficacy was made with butyrate-producing bacteria, i.e., mainly the genera *Ruminococcus*, *Clostridium*, *Eubacterium*, and *Coprococcus* [205]. *C. butyricum* in combination with *Lim. reuteri* effectively attenuated cDDP-associated renal damage; moreover, when given concomitantly with cDDP, it enhanced chemosensitivity in cellular models as a potential histone deacetylase inhibitor [143,206].

In terms of RT, the microbiome is an important player in local control, as well as a systemic therapy response, regardless of the site of RT [62,146,192,193,207,208,209,210,211,212]. Clinical studies on rectal cancer convincingly report that the specific gut microbiome is associated with CRT response in relation to reaching complete remission or not [146,192,193,207]. Microbial alterations were also linked to radiation-induced adverse events in patients with prostate, gynecological, rectal, and other cancers [147,148,208,209,210,211,212,213,214,215,216,217,218,219,220,221,222,223,224,225,226,227,228,229,230,231,232,233,234,235,236,237]. Focusing on the oral microbiome and HNSCC, the depth of knowledge is far shallower. RT of the head and neck region results in oral microbiome alterations; however, its pre-RT constitution is associated with the incidence and severity of RIOM [16,106,107,108]. Interestingly, so far, no specific microbial taxa have been linked to either radiosensitivity or radioresistance in HNSCC. The only clinical studies that demonstrated a significant effect on patients’ prognosis and treatment response reported that broad-spectrum antibiotics administered before or during CRT were linked with shorter time to recurrence and shorter survival [108,110]. These findings correlate with pre-clinical data investigating the relationship between dysbiosis, butyrate acid, and RT. Uribe-Herranz and colleagues investigated the impact of antibiotics on RT efficacy in a melanoma and lung/cervical model in C57/BL6 mice. Mice were stratified into three groups: (a) vancomycin-treated (covers butyrate-producing bacteria), (b) neomycin/metronidazole-treated (does not affect butyrate-producing bacteria), and (c) vancomycin-treated mice who had per oral intake of butyrate. Vancomycin-treated mice had a better response to RT; however, this effect was abrogated in the group with a dietary intake of butyrate, and neomycin/metronidazole-treated mice exhibited no change in response to RT [223]. Whether these findings can be extrapolated to the HNSCC is a subject of discussion, as no direct evidence exists; however, it would be highly interesting to investigate, especially considering that butyrate was observed to have a chemosensitizing effect with cDDP on cellular lines [143,180].

Only limited data are available regarding the role of the human microbiome and its metabolites on ICI efficacy in R/M HNSCC. Bari et al. reported a significantly higher abundance of *Eu. oxidoreductans*, *Bact. uniformis*, and *Ruminococcus* spp. in the ICI responders [187]. Interestingly, these bacteria are closely associated with butyrate production [205,237], which is consistent with preclinical findings that butyrate enhances ICI-treatment efficacy via activation of CD8+ [172,180,238,239,240]. Across the immunotherapy of solid tumors, it is becoming a generally accepted fact in the scientific community that the host microbiome can detrimentally affect ICI treatment efficacy. Experiments with fecal transplantation (FMT) showed that the response to ICI treatment can be modified via a change in gut microbiome. Routy et al. [171] hypothesized that abnormal gut microbiome composition is associated with primary resistance to the ICIs. Moreover, they described how mice react to ICI treatment in several situations: (a) germ-free mice did not respond to ICIs; (b) germ-free mice who received FMT from a human responder to ICIs responded to the therapy; (c) germ-free mice who received FMT from a human non-responder to ICIs did not respond to the therapy; and (d) germ-free mice who received FMT from a human non-responder to ICIs, but had per orally administered *A. muciniphila*, responded to the ICIs [171]. Bernal et al. observed that *A. muciniphila* was significantly less abundant in the stool of oropharyngeal SCC patients [217]. Other authors suggested that the higher abundance of *A. muciniphila* can be related to better treatment outcomes with ICIs [172,173,180]. Unfortunately, the situation is not that simple. Based on the decreased representation of *A. muciniphila* in HPV+ HNSCCs, we could expect worse treatment efficacy of ICIs in HPV-associated HNSCCs; however, the situation is reversed, i.e., the efficacy of ICIs is in HPV-associated rather than in HPV-negative HNSCCs (1.29× higher response rate, median OS 11.5 vs. 6.3 months) [241]. So far, the only clinical trial investigating the association between the oral microbiome and ICI treatment outcomes (specifically with nivolumab) was conducted by Farris and colleagues, and they failed to report any significant association [188].

The effects of the oral microbiota on treatment outcomes in HNSCC are summarized in Appendix A.

### 4.3. Probiotics Are Potent Agents in Anti-Cancer Therapy and Toxicity Modulation in HNSCC

Probiotic administration is intensively studied in efforts to enhance anti-cancer treatment outcomes and/or to reduce the severity of treatment toxicities. Lactobacilli, *Bifidobacterium* spp., *Lactococcus* spp., and *Enterococcus* spp. appear to be promising probiotic taxa with such activities [242,243]. The effect of probiotics on the alleviation of RT-induced adverse toxicity was mostly studied in patients undergoing pelvic RT; however, the evidence from clinical trials is still limited [147,148,231,232]. Several authors reported positive effects of *Lactobacillus*-based probiotics on RT-induced diarrhea [215,226,228,234].

So far, limited evidence regarding the probiotic effect in HNSCC has been published; however, similar results have been reported. Probiotic strains of *Lactobacillus* spp., *Lactococcus* spp., and *Bifidobacterium* spp. were all observed to reduce RT- and/or CHT-induced mucositis in HNSCC treatment [110,111,112,113,114]. Interestingly, a decrease in bifidobacterial representation was observed after RT [222]. It was hypothesized that, thanks to bifidobacterial anti-cancer effects and their beneficial effect on the oral microbiome, they could be an interesting agent for concomitant and/or adjuvant administration to HNSCC patients [195,196,197,198]. Similar results were observed in HNO97 cellular cultures with *L. acidophilus* and *S. mutans*, promoting further clinical research [244].

Based on the results of patients who had been pre-treated with probiotics and then received surgery for CRC, it was theorized that proper nutritional preparations and probiotics administration can improve results and/or decrease surgery-related toxicity in HNSCC patients undergoing radical surgical treatment; however, no prospective trials have been completed [245,246]. Limited data on immunonutrition in HNSCC are available, mostly regarding nutritional support [247]. Nevertheless, oral and/or enteral administration of probiotics is safe and helpful in patients who have undergone radical surgery for HNSCC. Gunduz et al. observed lower complications in patients after the HNSCC surgery, including a decreased rate of post-operative sepsis [248].

The effect of probiotics on HNSCC treatment is summarized in Appendix A.

## 5. Conclusions

The microbiome plays an important role in cancer development, progression, and treatment response across solid tumors, including HNSCC. Recent studies highlighted significant discrepancies in microbial profiles of healthy individuals and of those with oral potentially malignant disorders (e.g., OPMD) and/or HNSCC, suggesting the microbiome’s potential as a biomarker and therapeutic target. The association between oral dysbiosis and various malignancies initiated efforts to identify specific microbial agents tied to cancer development and/or progression.

Due to the significant heterogeneity of HNSCCs and the vast diversity of the oral microbiome, inconsistent associations of increased/decreased microbial diversity and/or relative/absolute abundance of specific taxa in the salivary microbiome with HNSCC were observed. In HNSCC, cancer-protective properties are commonly attributed to *Neisseria* spp.; conversely, *F. nucleatum*, *Lactobacillus* spp., *S. anginosus*, and *P. gingivalis* exhibit cancerogenic effects. Interestingly, a high abundance of *F. nucleatum* in the oral microbiome appears to be associated with better prognosis, which is inconsistent with its negative prognostic role in other cancers.

The high inconsistency among reported data can be possibly explained by several theories; however, conclusive evidence is still missing. Microbial diversity and ranging microbial representation in HNSCC patients’ samples can be attributed to the interindividual diversity, which can be observed in healthy controls, as well as to incoherent study designs, including different samples, collection methods, and patients’ characteristics (e.g., diet, smoking habits, alcohol consumption, etc.). Further controlled studies focusing on the covariant patients’ characteristics are necessary to establish causal associations.

Almost no research data suggest that the oral microbiome would be of a similar importance compared to the gut microbiome in terms of affecting the systemic therapy response (meaning CHT and ICIs therapy) and/or its toxicity.

Surgical treatment and (C)RT led to notable changes in the oral microbiome. (C)RT of HNSCCs may lead to interrupted salivary flow via collateral radiation damage to salivary glands. Decreased salivation and xerostomia results in significant microbial changes, e.g., a higher relative abundance of the genera *Streptococcus*, *Staphylococcus*, and *Lactobacillus*, and a lower relative abundance of the genera *Haemophilus*, *Neisseria*, and *Fusobacterium*, which ultimately affect patients’ QoL and their oral cavity health. Moreover, pre-existing dysbiosis, especially the representation of some Gram-negative bacteria (e.g., *Fusobacterium* spp., *Haemophilus* spp., *Porphyromonas* spp., and *Eikenella* spp.), was observed to impact both the incidence and severity of adverse events such as RIOM and xerostomia. Interventions with *Bifidobacterium-* and/or *Lactobacillus*-based probiotics and some prebiotics have shown potential in alleviating the severity and incidence of RIOM.

Specific microbiota can impact the pharmacodynamics of chemotherapeutics like cDDP and 5-FU, which are routinely used in HNSCC treatment regimens, thus affecting the treatment efficacy and toxicity. *Lactobacillus* spp. and *A. muciniphila* were observed to enhance cDDP-treatment response in animal models, whereas others, like *F. nucleatum*, may contribute to chemoresistance and treatment failure. *Lactobacillus*-based probiotics showed decreased 5-FU-related toxicity in rats. So far, no research has evaluated the impact of the oral microbiome on the ICI response rate and efficacy in HNSCC treatment. The gut microbiome appears to play a pivotal role in predicting the ICI treatment efficacy and even toxicity, whereas the oral microbiome has not shown any significant impact on ICI efficacy.

Microbiome alterations through pre-/pro-/synbiotics to achieve a better treatment response and/or to reduce the treatment toxicities have been demonstrated on several levels, e.g., a reduction in acute RT-induced mucositis in HNSCC RT, FMT from ICI responders to non-responders to overcome primary resistance to ICIs, etc. So far in HNSCCs, *Lactobacillus*- and/or *Bifidobacterium*-based probiotics have been shown to decrease RIOM incidence and severity. It also appears that probiotics could serve as potential adjuncts to conventional therapies in improving treatment outcomes and/or reducing treatment toxicity.

## Figures and Tables

**Figure 1 cancers-17-02238-f001:**
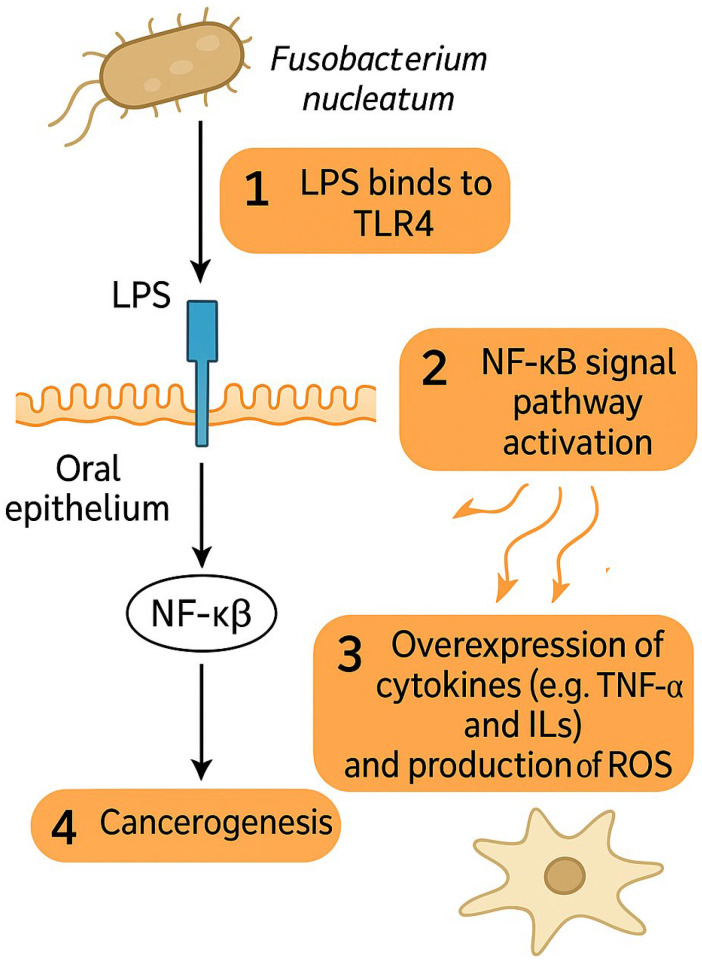
*F. nucleatum* promotes cancerogenesis via interaction of LPS with TLR4 on epithelial cells, which leads to overexpression of cytokines and ROS via NF-κB signal pathway activation. Abbreviations: LPS—lipopolysaccharide, TLR4—Toll-like receptor 4, NF-κB—Nuclear factor kappa-light-chain-enhancer of activated B cells, TNF-α—tumor necrosis factor alpha, ROS—reactive oxygen species.

**Figure 2 cancers-17-02238-f002:**
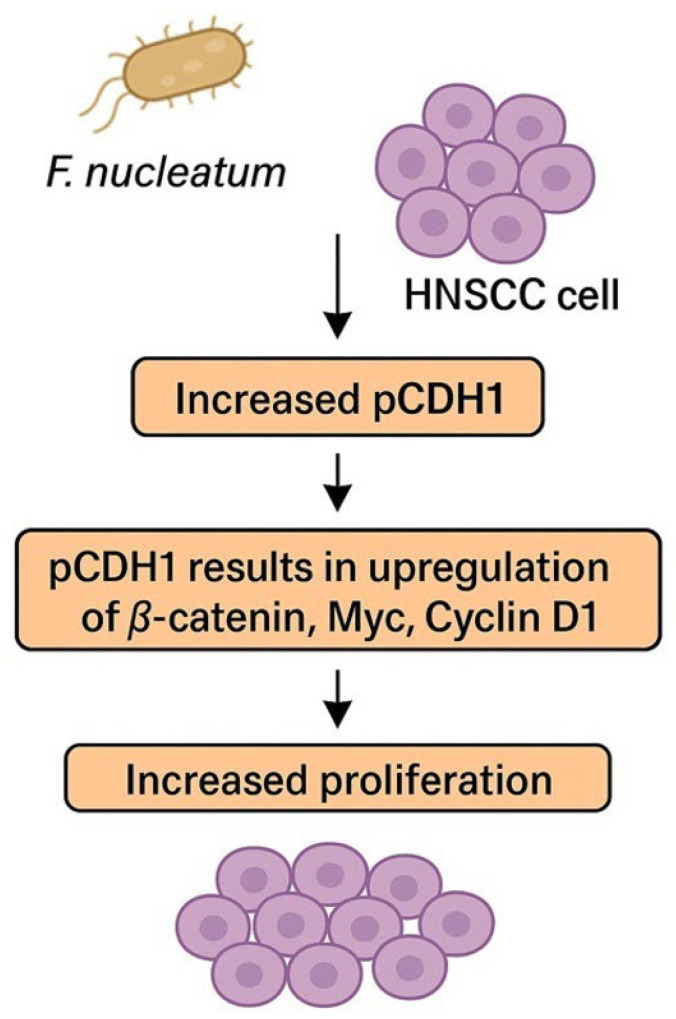
*F. nucleatum* promotes cancer cell proliferation via phosphorylation of CDH1. Abbreviations: HNSCC—head and neck squamous cell carcinoma, pCDH1—phosphorylated cadherin 1.

**Figure 3 cancers-17-02238-f003:**
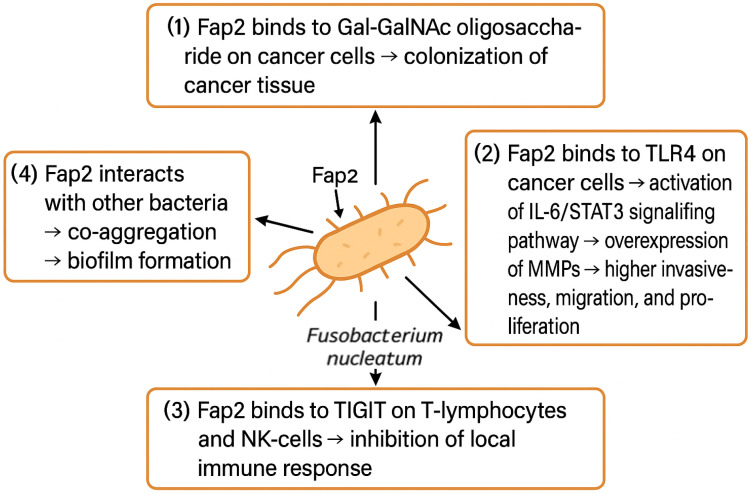
*F. nucleatum* promotes OSCC tissue colonization, proliferation, invasiveness, and migration and suppresses local immune response. Abbreviations: Gal-GalNAc—galactose- and *N*-acetyl-D-galactosamine, TLR4—Toll-like receptor 4, IL-6—interleukin 6, STAT3—signal transducer and activator of transcription 3, MMPs—metalloproteinases, TIGIT—T-cell immunoreceptor with Ig and ITIM domain, NK-cell—natural killer cell.

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
