# Peer review of "Role of Human Microbiome in Development and Management of Head and Neck Squamous Cell Carcinoma"

_cancers, 2025, doi:10.3390/cancers17132238_

Round 1

Reviewer 1 Report (Previous Reviewer 1)

Comments and Suggestions for Authors

The manuscript has been improved; however, it requires further revision before a decision can be made. These track changes make it very difficult to understand which parts have been added or which parts have been deleted. A colorful font or highlighting is much better.

  • I highly recommend that the authors draw a graphical abstract at the end of the introduction.
  • This manuscript does not have a methodology part (Materials and Methods). Moreover, I recommend cutting and moving the appendix into the supplementary and marking it as table S1-S4.
  • I would like to ask to enrich the result in the abstract with numerical results.
  • What does radical treatment mean? (line 321-page 8 of 49) Is it related to drugs that increase the ROS level?
  • In section 2.1 (line 116-page 3), the authors should proceed with molecular pathways (mechanisms). Head and neck cancer (or any type of cancer) has genetic, epigenetic profiles. If any agent (microbe, chemical agent...) changes these profiles can lead to irregulating in the cell’s function, which could lead to cancer development. In the microbial aspect, each type of microbe may have a distinct pathway. It is important to represent the pathways by each type of microbe that lead to head and neck cancer (or other cancers). The authors here only report on previous researchers.

Author Response

Dear reviewer, thank you for taking the time to review this manuscript. Please find the detailed responses below. 

Comment 1: I highly recommend that the authors draw a graphical abstract at the end of the introduction.

  • As for the narrative review, co-authors and I find it difficult to think of a graphical abstract to summarize our manuscript, therefore, we decided not to include this in the final version. 

Comment 2: This manuscript does not have a methodology part (Materials and Methods). Moreover, I recommend cutting and moving the appendix into the supplementary and marking it as table S1-S4.

  • The methodology part was committed in the previous revision as one of the Editors deemed it unnecessary in a narrative review. The methodology part was gained and included in the final version of the manuscript. The tables from the appendix were moved into the supplementary materials. 

Comment 3: I would like to ask to enrich the result in the abstract with numerical results.

  • Neither I nor my co-authers understood this comment, therefore no revision was made. 

Comment 4: What does radical treatment mean? (line 321-page 8 of 49) Is it related to drugs that increase the ROS level?

  • In oncology, radical treatment is generally used to indicate curative intent of the treatment. For example, radical chemoradiotherapy is a type of oncological treatment combining radiotherapy and chemotherapy to cure a patient, unlike palliative care.

Comment 5: In section 2.1 (line 116-page 3), the authors should proceed with molecular pathways (mechanisms). Head and neck cancer (or any type of cancer) has genetic, epigenetic profiles. If any agent (microbe, chemical agent...) changes these profiles can lead to irregulating in the cell’s function, which could lead to cancer development. In the microbial aspect, each type of microbe may have a distinct pathway. It is important to represent the pathways by each type of microbe that lead to head and neck cancer (or other cancers). The authors here only report on previous researchers.

  • Considering the variety and immense number of possible bacterial agents participating in the head and neck cancer genesis, detailed description of how these agents affect molecular pathways grossly exceeds the aims and possibilities of this review. As one of the key players in cancerogenesis, we focused on the molecular background of F nucleatum, which has been described in detail. 

Thank you for giving us the opportunity to submit a revised draft of the manuscript. We have incorporated most of the suggestions made by the reviewers. Those changes are highlighted within the manuscript. 

Yours sincerely

Martin Palkovsky

Reviewer 2 Report (Previous Reviewer 3)

Comments and Suggestions for Authors

This well-executed narrative review explores the clinically significant relationship between the human microbiome and head and neck squamous cell carcinoma (HNSCC). The authors skillfully integrate diverse evidence from meta-analyses, preclinical studies, and contemporary clinical trials to provide a sophisticated analysis of how oral and gut microbiota modulate carcinogenesis, therapeutic efficacy, and treatment-related toxicities in HNSCC patients.

Author Response

Dear reviewer, 

Thank you for taking the time to review our manuscript. We appreciate you considering our work.    We have carefully reviewed your feedback and note that you did not provide any specific suggestions for improvement. While we understand that you may have found the manuscript satisfactory as is, we made subtle changes based on the recommendation of other reviewers.    We added a methodology part. Tables from the appendix were moved into the supplementary materials. All changes are highlighted in the revised manuscript.   We are committed to improving the quality of our work and welcome any further insights you might have.    Yours sincerely Martin Palkovsky

Round 2

Reviewer 1 Report (Previous Reviewer 1)

Comments and Suggestions for Authors

The quality of the manuscript is okay, but the authors did not provide a graphical abstract and also omitted the methodology section. 

This manuscript is a resubmission of an earlier submission. The following is a list of the peer review reports and author responses from that submission.

Round 1

Reviewer 1 Report

Comments and Suggestions for Authors
  • I suggest the authors expand the results part in the abstract, especially by adding numerical results. The current result is a general fact about the microbiome.
  • I strongly suggest that the authors add a graphical abstract at the end of the introduction, e.g., various procedures that the human microbiome increases/decreases the risk of head and neck squamous cell carcinoma.
  • The structure of this review manuscript is not good. Methods are used for research articles. If these authors want to describe their method, is OK but the methods should be completely described.
  • The structure of the results is also not good. Each section has its own conclusion, and section 4 also has a conclusion. The description volume in the conclusion is larger than the main results!!!. The authors must move the tables to each section. It is also good to list notable microbes that have a role in the prevention of HNSCC.
  • It is good that the authors discuss the effects of using antibiotics to treat infection on microbiome and HNSCC development/prevention.
  • The authors did not use figures at all. They could use figures to transfer information to readers.

Reviewer 2 Report

Comments and Suggestions for Authors

The manuscript provides a comprehensive review of the microbiome's role in HNSCC. It has the potential to make a valuable contribution to microbiome-associate cancer research and treatment. following suggestions aim to enhance the manuscript: to clarify the microbiome-associate molecular mechanisms for better understanding of causality, to integrate the clinical impact of   microbiome-based therapies, and to ensure balanced citation of sources. 

Comments on the Quality of English Language

The manuscript is grammatically sound and maintains a professional, scientific tone, but readability can be improved by simplify lengthy sentences and eliminating redundant phrases. 

Reviewer 3 Report

Comments and Suggestions for Authors

This manuscript addresses the relationship between the human microbiome and head and neck squamous cell carcinoma (HNSCC), reflecting recent research trends in this emerging field. The study provides a comprehensive literature review, incorporating systematic reviews, meta-analyses, and recent studies, making it a valuable resource for understanding the role of the microbiome in cancer development and treatment. Furthermore, the paper effectively analyzes the impact of the microbiome on treatment response and toxicity, which is an important aspect of precision oncology. The structure of the manuscript is clear, following a logical progression from microbiome alterations - treatment response - clinical applications, making it easy to follow.

However, there are several areas that require improvement:

  1. The manuscript does not sufficiently explain the underlying molecular mechanisms by which the microbiome influences cancer progression and treatment response.
  2. Include diagrams or graphical summaries illustrating these mechanisms, such as how Fusobacterium nucleatum modulates immune evasion, epithelial-mesenchymal transition (EMT), and DNA repair pathways.
  3. The results regarding microbial diversity in HNSCC are inconsistent, with some studies reporting an increase, while others report a decrease.
  4. A more detailed discussion on the variability of findings is needed, considering differences in sampling methods (saliva, tumor tissue, plaque, etc.), patient characteristics (smoking, diet), and study designs.
  5. The study primarily highlights associations between microbiome alterations and HNSCC but does not adequately discuss whether these relationships are causal.
  6. Consider using frameworks such as Hill’s criteria for causation or discussing experimental approaches (e.g., animal models, mechanistic studies) to strengthen the argument.
  7. The presentation of conclusions is inconsistent—some results sections include conclusions, while others do not, and Section 4 presents a separate conclusion again, leading to redundancy.
  8. The title formatting and use of abbreviations are not consistent throughout the manuscript.
  9. Ensure a unified structure, keeping conclusions only in the dedicated "Conclusion" section and maintaining consistent abbreviation usage throughout the text.